# Observer-Based Finite-Time Dynamic Encirclement for Multi-ASV Systems Using Time-Varying Sliding Mode Control

Jiahui Zhang
*School of Navigation*
*Wuhan University of Technology*
Wuhan, China
jiahuizhang@whut.edu.cn

Yue Yang*
*School of Navigation*
*Wuhan University of Technology*
Wuhan, China
yueyang@ieee.org

Kezhong Liu
*School of Navigation*
*Wuhan University of Technology*
Wuhan, China
kzliu@whut.edu.cn

Xiaochen Li
*Tianjin Research Institute for Water*
*Transport Engineering, M.O.T.*
Tianjin, China
Lixch@tiwte.ac.cn

*Abstract*—Dynamic encirclement is a strategy employed to maintain ongoing monitoring or neutralize a target by restricting its range of movements. This paper proposes a novel approach to handle finite-time dynamic encirclement problems for multi-autonomous surface vehicle (ASV) systems. To be consistent with the actual task scenarios, each ASV faces challenges such as model uncertainties, unavailable velocities, saturated inputs, and external disturbances. Specifically, a fuzzy logic system (FLS) is utilized to approximate the nonlinear dynamics, and an adaptive fuzzy state observer is designed to estimate the unavailable velocities of the multi-ASV system. Next, a time-varying sliding mode controller is developed based on a time-varying formation function. In order to prevent singularity, the system is divided into two zones. A terminal sliding mode controller and a linear auxiliary sliding mode controller are assigned to each zone, respectively. Then, by applying finite-time theory, the dynamic encirclement issues of multi-ASV systems can be addressed within a finite time. Finally, the effectiveness of the obtained results can be verified by the Lyapunov theory and simulation examples presented.

*Index Terms*—Finite-time dynamic encirclement, Time-varying sliding mode control, Autonomous surface vehicle (ASV), Adaptive fuzzy state observer

## I. INTRODUCTION

With advancements in control engineering, communication capabilities, and maritime technology, formation control of multi-autonomous surface vehicle (ASV) systems has garnered considerable attention in recent years. Up to now, applications in both the civilian and military domains employ formation control of multi-ASV systems., including exploring marine resources, formation escort, executing transportation tasks, and so on [1] [2].

In contrast to individual ASV, the multi-ASV system offers advantages such as improved performance, task execution efficiency, and reliability [3] [4]. Thus far, numerous formation control methods have been presented in multi-ASV systems, such as the artificial potential field method [5], the virtual structure method [6], the leader-follower method [7] [8], and so forth. In addition to these approaches, the combined leader-follower within the graph theory approach has gained popularity in recent years because the followers in the formation can make their own decisions based on local information exchanges. Although the leader-follower approach is desirable, the formation configuration is fixed in the aforementioned works. The flexibility of formation cannot satisfy the requirements in complicated scenarios, such as scenarios with static or moving obstacles, narrow channels, and so on.

In order to overcome this limitation, many studies are proposed to improve the diversity of formation configurations. In [9], time-varying formation control is first introduced in the field of unmanned aerial vehicles (UAVs). Unlike the previously studies [3] [5] [7] [8], formation patterns can be modified using time-varying formation functions. In [10], a time-varying formation controller based on adaptive NNs is designed for nonlinear multi-agent systems (MASs). However, the research in [9] [10] was limited to UAVs and MASs. The relevant results cannot be directly employed with multi-ASV systems because of the external disturbances and the strong nonlinearity of ASVs [11].

Flexible formation configurations are necessary for multi-ASV systems to complete complicated working scenarios, such as crossing narrow channels [12], avoiding obstacles [13], executing dynamic encirclement tasks [14], and so on.

This paper is supported by the National Key Research and Development Program of China (2023YFB2603800), the National Natural Science Foundation of China (NSFC) under Grant No. 52031009, and the Fundamental Research Funds for the Central Universities (WUT: 2024IVA046). (Corresponding author: Yue Yang).

Specifically, dynamic encirclement is a significant strategy that involves surrounding a target and employing encircling motion to limit its movement, ultimately leading to its capture. For a multi-ASV system, encircling and limiting its movement enables a follower ASV team to monitor the target in real-time. In [15], a distributed controller is developed to enable the agents to move around the target. An estimator is designed for each agent to identify the target by utilizing the bearing measurements of both the agent and its surrounding agents. To solve dynamic encirclement problems for multi-ASV systems, the authors in [14] propose a time-varying formation controller and feasible conditions. In addition, nonlinear dynamics and actuator saturation are considered. In [16], an integrated distributed guidance and a model-free control method are presented to achieve target enclosing control. The cooperative moving target enclosing and tracking problem is addressed in [17] by a dynamic distributed control law for networked, unicycle-type vehicles moving at constant linear velocities. On the one hand, the above methods rely on the position and velocity state information. However, most ASVs are not capable of obtaining velocity state information in real-time. On the other hand, the methods in [14] [16] [17] cannot guarantee that the controlled object will achieve the control objective within a finite time.

Motivated by the above discussions, this paper develops a distributed time-varying formation sliding mode controller to address finite-time dynamic encirclement issues for the multi-ASV with unavailable velocity state information, saturated inputs, and external disturbances. First, to estimate the unavailable velocity state information by using position and heading state information, an adaptive fuzzy state observer is proposed based on the approximation of nonlinear dynamics by utilizing the fuzzy logic system (FLS) [18] [19]. Second, a novel distributed controller for the ASV team is designed by combining finite-time theory with the time-varying sliding mode control approach. Third, to be consistent with actual task scenarios, saturated inputs are also considered due to the power limitation of actuators. Finally, simulation results are presented to demonstrate the efficiency of the proposed approach.

The primary contributions of this paper are listed as follows:

1) To address finite-time dynamic encirclement issues for multi-ASV systems, a time-varying formation sliding mode controller is developed based on a predefined time-varying formation function. The formation patterns can be adjusted by modifying this function.

2) By integrating the finite-time stability theory with the time-varying formation sliding mode control approach, the ASV team can achieve time-varying formation while dynamically encircling the target within a finite time.

3) To be consistent with the actual situation, each ASV faces unavailable velocities, saturated inputs, and external disturbances. The nonlinear dynamics are approximated by utilizing a FLS. After that, an adaptive fuzzy state observer is designed to estimate the velocity state information from the position and heading state information. Additionally, the control inputs are restricted within a specific range by applying a saturation function.

The rest of this paper is structured as follows: In Section II, some preliminaries are presented, and a dynamic encirclement movement description is given. In Section III, an adaptive fuzzy state observer-based time-varying sliding mode controller is developed. In addition, the effectiveness of the developed controller is analyzed. In Section IV, a series of simulation results and discussions are presented. The relevant conclusions are introduced in Section V.

## II. PRELIMINARIES

In this section, the finite-time stability theory is described. The mathematical models of a ASV team and a target are provided.

### A. Basic Graph Theory

Denoted a directed graph as $G = \{N, E, A\}$ to describe the topology connection of a multi-ASV system communication network. $N = \{1, 2, ..., n\}$ is the set of nodes to represent the $n$ ASVs. $E \subseteq N \times N$ is the set of edges to represent the information transfer among all neighboring ASVs. $A = [a_{ij}] \in R^{n \times n}$ is the adjacency matrix to represent the communication quality between two neighboring ASVs. Besides, the neighborhood set can be defined as $\Theta_i = \{j \in N | (i, j) \in E\}$. If and only if the information can be transformed from the $j$th ASV to the $i$th ASV, the adjacency elements $a_{ij} = 1$ and $a_{ij} = 0$ for other cases. For a directed graph, the in-degree matrix is denoted as $D = diag\{d_1, d_2, ..., d_n\}$ and the Laplacian matrix $\ell$ can be expressed as

$$\ell = [\ell_{ij}] = D - A. \tag{1}$$

*Assumption 1:* Consider the weakly linked network composed of $N + 1$ ASVs. It means that there exists at least one directed path from the target ASV to any ASVs in the ASV team. Note that there is no directed path from any ASV in the ASV team to the target ASV. In addition, the communication relationship between all ASVs of the ASV team in remains fixed.

$L = \{l_{ij}\}$ is the normalized directed Laplacian matrix that is expressed as

$$l_{ij} = \begin{cases} 1, & i = j, i \neq 0 \\ \ell_{ij}/\ell_{ii}, & i \neq j, i \neq 0 \\ 0, & otherwise, \end{cases} \tag{2}$$

where $l_{ij} \in R^{(N+1) \times (N+1)}$, $i = 1, 2, ..., n$, $j = 0, 1, ..., n$. $\ell_{ii} = \sum_{i \neq j} a_{ij}$ and $\ell_{ij} = -a_{ij}$ for $i \neq j$. The subscripts $1, 2, ..., n$ are used to represent the ASV in an ASV team, and the subscript 0 is used to represent the target ASV.

*Remark 1:* Since the target ASV cannot transform any information to the ASV team, the Laplacian matrix $L$ (2) can be expressed as

$$L = \begin{bmatrix} 0 & 0_{1 \times n} \\ l_1 & l_2 \end{bmatrix}, \tag{3}$$

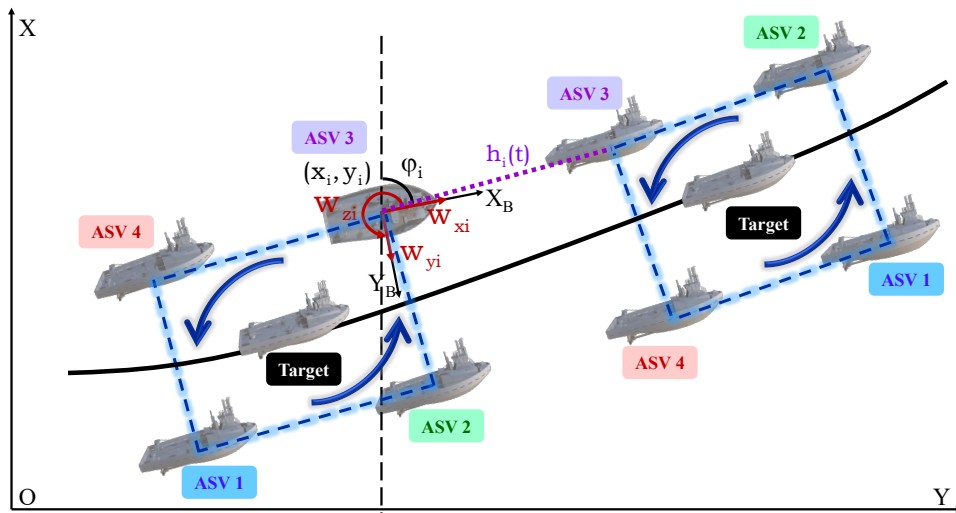

Fig. 1. A dynamic encirclement movement description.

where

$$l_1 = \begin{bmatrix} l_{10} \\ \vdots \\ l_{n0} \end{bmatrix}, l_2 = \begin{bmatrix} l_{11} & \cdots & l_{1n} \\ \vdots & \ddots & \vdots \\ l_{n1} & \cdots & l_{nn} \end{bmatrix}. \quad (4)$$

### B. Finite-Time Stability Theory

Let $\partial\Omega$, $\bar{\Omega}$, and $\mathring{\Omega}$ represent the boundary, the closure, and the interior of the set $\Omega \in R^{n\times1}$, respectively. $\|\cdot\|_\infty$ is the norm of $\ell_\infty$, $V'(x)$ is denoted as the Fréchet derivative of $V$ at $x$.

Then, the nonlinear dynamical system is considered as

$$\dot{x}(t) = f(x(t)), x(0) = x_0, t \geq 0 \quad (5)$$

where $x(t) \in \Omega \subseteq R^{n\times1}, t \geq 0$ represents the state vector. The set $\Omega$ is open, $0 \in \Omega$, $f(0) = 0$, and the function $f(\cdot)$ maintain continuity over $\Omega$. With the initial condition $x_0 \in \Omega$, we can instead represent the solution to (5) by $q^{x_0}(t)$, $t>0$, or by $q(x_0, t)$, $t \geq 0$. Next, the following lemmas are necessary for system analysis.

*Lemma 1 ( [20]):* Given (5), let $\Omega_0 \subset \Omega$ be a positively invariant set. $\Omega_0$ is finite-time stable if it has an open neighborhood $Z \subset \Omega$ and a settling-time function $T : Z\backslash\Omega_0 \to (0,\infty)$, such that the following statements hold.

1) Lyapunov stability: For any open neighborhood $\Gamma_1 \subseteq Z$ containing $\Omega_0$, there exists a smaller open neighborhood $\Gamma_2 \subseteq \Gamma_1$ of $\Omega_0$ such that for every $x \in \Gamma_2\backslash\Omega_0$, the trajectory $q(x,t)$ remains within $\Gamma_1$ for all $t \in [0, T(x)]$.

2) Finite-time convergence: $q(x,t)$ is defined on $[0, T(x))$, $q(x,t) \in Z\backslash\Omega_0$ for all $t \in [0, T(x))$, and $\lim_{t\to T(x)} dist(r(t,x), \Omega_0) = 0$ for each $x \in Z\backslash\Omega_0$.

Consequently, if $\Omega_0$ is finite-time stable with $Z = \Omega = R^n$, then it is globally finite-time stable.

*Lemma 2 ( [8]):* With respect to (5), let $\Omega_0 \subset \Omega$ be an invariant set regard. $\Omega_0$ is finite-time stable if there exists a continuously differentiable function $V : \Omega \to R$, along with positive real numbers $c$ and $d \in (0,1)$, such that $V(x) = 0$ for $x \in \Omega_0$ and $V(x)>0$ for $x \in \Omega\backslash\Omega_0$, and

$$V'(x)f(x) \leq -c(V(x))^d, x \in \Omega. \quad (6)$$

Additionally, if $T : Z \to [0,\infty)$ represents the setting-time function and $Z$ is defined as in Lemma 1, then

$$T(x_0) \leq \frac{1}{c(1-d)}(V(x_0))^{1-d}, x_0 \in Z \quad (7)$$

and $T(\cdot)$ is continuous on $Z$. Furthermore, $\Omega = R^n$. If (5) holds on $R^n$ and $V(\cdot)$ is radially unbounded, then $\Omega_0$ is globally finite-time stable.

This lemma outlines the sufficient conditions for a set that is invariant to be finite-time stable.

### C. The ASV Model formulation

A dynamic encirclement movement description is presented in Fig 1. The kinematics and dynamics model of the ASV is given as follows:

$$\begin{aligned} \dot{\eta}(t) &= R(\psi)w(t) \\ I\dot{w}(t) &= -C'(w)w(t) - D'(w)w(t) - \delta'(t) + \tau'(t) \end{aligned} \quad (8)$$

where $\eta = [x, y, \psi]^T$ represents the position and heading state of the ASV in the earth-fixed reference coordinate, $w = [w_x, w_y, w_z]^T$ denotes the velocity state of the ASV in the body-fixed reference coordinate, $\tau' = [F_x, F_y, T_z]^T$ denotes the control input, $\delta' = [\delta'_x, \delta'_y, \delta'_z]^T$ denotes the external disturbance, $R(\psi)$ represents the rotation matrix with $R^T(\psi) = R^{-1}(\psi)$, it can be expressed as follows:

$$R(\psi) = \begin{bmatrix} \cos(\psi) & -\sin(\psi) & 0 \\ \sin(\psi) & \cos(\psi) & 0 \\ 0 & 0 & 1 \end{bmatrix}.$$

The system inertial matrix, denoted by $I$, is mostly based on the mass of the ASV.

$$I = I^T = \begin{bmatrix} m_{11} & 0 & 0 \\ 0 & m_{22} & m_{23} \\ 0 & m_{32} & m_{33} \end{bmatrix}.$$

The nonlinearity of the ASV dynamics model determines both the Coriolis matrix, defined as $C(w)^T = C(w)^{-1}$, and the hydrodynamic damping matrix, represented by $D(w)$. The following formulations describe $C(w)$ and $D(w)$:

$$C(w) = \begin{bmatrix} 0 & 0 & c_{13} \\ 0 & 0 & c_{23} \\ -c_{13} & -c_{23} & 0 \end{bmatrix}$$

where $c_{23} = m_{11}w_x$, $c_{13} = -m_{22}w_y - 1/2(m_{23} + m_{32})v_z$. The hydrodynamic parameters are defined as $X_{(\cdot)}, Y_{(\cdot)}, J_{(\cdot)}$.

$$D(w) = \begin{bmatrix} d_{11} & 0 & 0 \\ 0 & d_{22} & d_{23} \\ 0 & d_{32} & d_{33} \end{bmatrix}$$

where

$$d_{11}(w) = -X_x - X_{|x|x} - X_{xxx}w_x^2,$$
$$d_{22}(w) = -Y_y - Y_{|y|y}|w_y| - Y_{|z|y}|w_z|,$$
$$d_{23}(w) = -Y_z - Y_{|y|z}|w_y| - Y_{|z|z}|w_z|,$$
$$d_{32}(w) = -J_y - J_{|y|y}|w_y| - J_{|z|y}|w_z|,$$
$$d_{33}(w) = -J_z - J_{|y|z}|w_y| - J_{|z|z}|w_z|.$$

With $R(\psi)w(t) = v(t)$, the dynamics model (8) of the ASV can be expressed as

$$\dot{\eta}(t) = v(t)$$
$$\dot{v}(t) = \delta(t) + \tau(t) + D(v)v(t) + C(v)v(t) \tag{9}$$

where

$$\delta(t) = -R(\psi)I^{-1}\delta'(t), \ \tau(t) = R(\psi)I^{-1}\tau'(t),$$
$$D(v) = \dot{R}(\psi)R^{-1}(\psi) - R(\psi)I^{-1}D'(R^{-1}(\psi)v)R^{-1}(\psi),$$
$$C(v) = -R(\psi)I^{-1}C'(R^{-1}(\psi)v)R^{-1}(\psi).$$

Let $C(v)v(t) + D(v)v(t) = f(\eta, v, t)$. Then, an ASV team consisting of $n$ ASVs with unavailable velocities and saturated inputs is considered. The dynamics of an individual ASV are expressed as

$$\dot{\eta}_i(t) = v_i(t)$$
$$\dot{v}_i(t) = f_i(\eta_i(t), v_i(t), t) + \delta_i(t) + \tau_{is}(t) \tag{10}$$
$$i = 1, 2, ..., n$$

where $v_i = [v_{xi}, v_{yi}, v_{zi}]^T$ denotes the unavailable velocity state vector, $\delta_i = [\delta_{xi}, \delta_{yi}, \delta_{zi}]^T$ denotes the external disturbances, $\tau_{is} = [\tau_x, \tau_y, \tau_z]^T$ represents the saturated control input and it can be described as

$$\tau_{is} = \begin{cases} \tau_{is} & |\tau_{is}| \le \zeta \\ sign(\tau_{is})\zeta & |\tau_{is}| > \zeta \end{cases} \tag{11}$$

where $\zeta$ is the saturation boundary.

As previously mentioned, the target is denoted with a subscript 0. The following describes the dynamics of the target ASV:

$$\dot{\eta}_0(t) = v_0(t)$$
$$\dot{v}_0(t) = f_0(\eta_0(t), v_0(t), t) \tag{12}$$

where $v_0$ denotes the velocity state, $\eta_0$ represents the position state, and $f_0$ denotes a limited unknown function.

## III. ADAPTIVE FUZZY STATE OBSERVER BASED SLIDING MODE CONTROLLER DESIGN

An adaptive fuzzy state observer-based sliding mode controller for multi-ASV systems is designed in this section. The effectiveness of the designed observer and controller is validated by utilizing the Lyapunov theory.

### A. Fuzzy logic system

*Lemma 3:* For any given continuous function $f(x)$ defined on a compact set $\nabla$. It can be approximated by a FLS with the approximation error $\sigma$:

$$\sup_{x \in \nabla} |f(x) - W^*\Phi(x)| \le \sigma \tag{13}$$

where $W^* = [W_1^*, W_2^*, ..., W_m^*]^T$ is the ideal weighted vector, and satisfies $\|W^*\| \le \bar{W}$ with $\bar{W} > 0$ being an unknown constant, $\sigma$ is assumed as a constant, the number of fuzzy rules is denoted as $m$.

$$\Phi_i(x) = \exp[\frac{-(x_i - \mu_i)^T(x_i - \mu_i)}{\phi_i^2}] \tag{14}$$

where $\Phi_i(x)$ is the Gaussian function, $\phi$ and $\mu_i$ are the width and center vector of the Gaussian function, respectively.

### B. Fuzzy State Observer Design

Based on the excellent approximation ability of the FLS, the state observer is designed by

$$\dot{\hat{\eta}}_i(t) = \hat{v}_i(t) + k_1(\eta_i(t) - \hat{\eta}_i(t))$$
$$\dot{\hat{v}}_i(t) = \hat{W}_i^T\Phi_i(\hat{v}_i) + k_2(\eta_i(t) - \hat{\eta}_i(t)) + \delta_i(t) + \tau_{is}(t) \tag{15}$$

where $\hat{\eta}_i = [\hat{x}_i, \hat{y}_i, \hat{\psi}_i]^T$ and $\hat{v}_i = [\hat{v}_{xi}, \hat{v}_{yi}, \hat{v}_{zi}]$ denote the position and velocity state information obtained by the designed state observer, $k_1$ and $k_2$ are positive gain matrices.

System (10) is equivalent to

$$\dot{\eta}_i(t) = v_i(t)$$
$$\dot{v}_i(t) = f(\hat{\eta}_i, \hat{v}_i, t) + \Delta f_i + \tau_{is}(t) + \delta_i(t) \tag{16}$$

where $f(\eta, v, t) = f(\hat{\eta}, \hat{v}, t) + \Delta f$.

Define the observer errors as $\tilde{\eta}_i(t) = \eta_i(t) - \hat{\eta}_i(t)$ and $\tilde{v}_i(t) = v_i(t) - \hat{v}_i(t)$. Then, the error dynamics of the $i$th ASV can be expressed as:

$$\dot{\tilde{\eta}}_i(t) = \tilde{v}(t) - k_1(\eta_i(t) - \hat{\eta}_i(t))$$
$$\dot{\tilde{v}}_i(t) = f(\hat{\eta}, \hat{v}, t) + \Delta f_i - \hat{W}_i^T\Phi_i(\hat{v}_i) - k_2(\eta_i(t) - \hat{\eta}_i(t)). \tag{17}$$

A fuzzy logic system (FLS) can be used to approximate the nonlinear function term $f(\hat{\eta}, \hat{w}, t)$ as follows

$$f(\hat{\eta}, \hat{v}, t) = W_i^{*T}\Phi_i(\hat{v}_i) + \sigma. \tag{18}$$

Taking (24) into (23), one has obtained

$$\dot{\tilde{\eta}}_i(t) = \tilde{v}(t) - k_1(\eta_i(t) - \hat{\eta}_i(t))$$
$$\dot{\tilde{v}}_i(t) = \tilde{W}_i^T\Phi_i(\hat{v}_i) + \sigma_i + \Delta f_i - k_2(\eta_i(t) - \hat{\eta}_i(t)). \tag{19}$$

Differential (15) yields

$$\ddot{\tilde{\eta}}_i(t) = \tau_{is}(t) + \hat{W}_i^T \Phi_i(\hat{v}_i) + \delta_i(t) + k_2 \tilde{\eta}_i(t) \\ + k_1 \tilde{v}(t) - k_1^2 \tilde{\eta}_i(t). \tag{20}$$

For sake of simplification, let $\hat{\theta}_i(t) = [\hat{\eta}_i^T(t), \hat{v}_i^T(t)]^T$, $\tilde{\theta}_i(t) = [\tilde{\eta}_i^T(t), \tilde{v}_i^T(t)]^T$, one has:

$$\dot{\tilde{\theta}}_i(t) = A_i \tilde{\theta}_i(t) + H_i(\tilde{W}_i^T \Phi_i(\hat{v}) + \sigma + \Delta f) \tag{21}$$

where $A_i = \begin{bmatrix} -k_1 & 1 \\ -k_2 & 0 \end{bmatrix}$, $H_i = [0,1]^T$.

In addition, if the components are chosen correctly, $A_i$ can be a stringent Hurwitz matrix. Currently, a positive definite matrix $P_i$ exists, making it

$$A_i^T P_i + P_i A_i = -Q_i \tag{22}$$

where $Q_i$ is a positive definite matrix.

*Theorem 1:* By a given adaptive fuzzy state observer, if the adaptive law satisfies

$$\dot{\hat{W}}_i = \Lambda_i \left[ \Phi_i(\hat{v}_i) \tilde{\theta}_i - \alpha_i \hat{W}_i \right] \tag{23}$$

where $\Lambda$ and $\alpha$ are both positive constants. The corresponding state observing errors $\tilde{\theta}_i$ are uniformly ultimately bounded (UUB).

*Proof:* The considered Lyapunov function is

$$V_{i0}(t) = \frac{1}{2} \tilde{\theta}_i^T P_i \tilde{\theta}_i + \frac{1}{2} \tilde{W}_i^T \Lambda_i^{-1} \tilde{W}_i. \tag{24}$$

Differential (24), one has get

$$\dot{V}_{i0}(t) = \frac{1}{2} \tilde{\theta}_i^T \left( A_i^T P_i + P_i A_i \right) \tilde{\theta}_i + \tilde{W}_i^T \Lambda_i^{-1} \dot{\hat{W}}_i \\ + \tilde{\theta}_i^T P_i H_i (\tilde{W}_i^T \Phi_i(\hat{v}_i) + \sigma_i + \Delta f_i) \\ = -\frac{1}{2} \tilde{\theta}_i^T Q_i \tilde{\theta}_i + \tilde{W}_i^T \Lambda_i^{-1} \dot{\hat{W}}_i \\ + \tilde{\theta}_i^T P_i H_i (\tilde{W}_i^T \Phi_i(\hat{v}_i) + \sigma_i + \Delta f_i). \tag{25}$$

By substituting the adaptive law (23), (25) can be rewritten as

$$\dot{V}_{i0}(t) = -\frac{1}{2} \tilde{\theta}_i^T Q_i \tilde{\theta}_i + \tilde{W}_i^T \Phi_i(\hat{v}_i) \tilde{\theta}_i - \alpha_i \tilde{W}_i^T \hat{W}_i \\ + \tilde{\theta}_i^T P_i H_i (\tilde{W}_i^T \Phi_i(\hat{v}_i) + \sigma_i + \Delta f_i). \tag{26}$$

Applying Young's inequality yields

$$\tilde{\theta}_i^T P_i H_i \sigma_i \leq \frac{1}{2} \left\| \tilde{\theta}_i \right\|^2 + \frac{1}{2} \| P_i \|^2 \| \sigma_i \|^2 \tag{27}$$

$$\tilde{\theta}_i^T P_i H_i \Delta f_i \leq \frac{1}{2} \left\| \tilde{\theta}_i \right\|^2 + \frac{1}{2} \| P_i \|^2 \| \Delta f_i \|^2 \tag{28}$$

$$-\alpha_i \tilde{W}_i^T \hat{W}_i \leq -\frac{\alpha_i \left\| \tilde{W}_i \right\|^2}{2} + \frac{\alpha_i \| W_i^* \|^2}{2} \tag{29}$$

$$\tilde{\theta}_i^T P_i H_i \tilde{W}_i^T \Phi_i(\hat{v}_i) \leq \frac{1}{2} \left\| \tilde{\theta}_i \right\|^2 + \frac{1}{2} \| P_i \|^2 \tilde{W}_i^T \tilde{W}_i \tag{30}$$

$$\tilde{W}_i^T \Phi_i(\hat{v}_i) \tilde{\theta}_i \leq \frac{1}{2} \left\| \tilde{W}_i \right\|^2 + \frac{1}{2} \left\| \tilde{\theta}_i \right\|^2 \tag{31}$$

Taking (27)-(31) into (26), it can be rewritten as follows

$$\dot{V}_{i0}(t) \leq -\lambda_0 \left\| \tilde{\theta}_i \right\|^2 + \frac{1}{2} (\| P_i \|^2 - \alpha_i + 1) \left\| \tilde{W}_i \right\|^2 + U_0 \tag{32}$$

where $\lambda_0 = \frac{1}{2}(\lambda_{min}(Q) - 5)$, $U_0 = \frac{\alpha_i \| W_i^* \|^2}{2} + \frac{1}{2} \| P_i \|^2 (\| \sigma_i \|^2 + \| \Delta f_i \|^2)$. ■

*Definition 1:* In this paper, the ASV team with unavailable velocities (15) is required to achieve time-varying formation while encircling the target. Therefore, the time-varying formation tracking errors are defined as

$$e_i(t) = \hat{\eta}_i(t) - h_i(t) - \eta_j(t) \\ (i = 1, 2, ..., n, j = 0, 1, ..., n, i \neq j, h_0(t) = 0) \tag{33}$$

where $h(t)$ represents the time-varying formation function of the ASV team corresponding to position and heading state.

Furthermore, with respect to the $i$th ASV, the generalized error state is defined as:

$$r_i(t) = \hat{\eta}_i(t) + \frac{\ell_{ij}}{\ell_{ii}} \sum [\hat{\eta}_j(t) + h_i(t) - h_j(t)] \\ (i = 1, 2, ..., n, j = 0, 1, ..., n, i \neq j, h_0(t) = 0). \tag{34}$$

*Lemma 4:* The normalized Laplacian matrix (3) and the generalized error state (34) are considered. If $r_i(t)$ converges to zero, $e_i(t)$ converges to zero. It means that dynamic encirclement is achieved.

Taking the second-order differentiation of (34) with respect to time yields the following equation:

$$\ddot{r}_i(t) = \ddot{\hat{\eta}}_i(t) + \frac{\ell_{ij}}{\ell_{ii}} \sum \left[ \ddot{\hat{\eta}}_j(t) + \ddot{h}_i(t) - \ddot{h}_j(t) \right] \\ = \tau_{is}(t) + \hat{W}_i^T \Phi_i(\hat{v}_i) + (k_2 - k_1^2) \tilde{\eta}_i(t) + k_1 \tilde{v}(t) \\ + \frac{\ell_{ij}}{\ell_{ii}} \sum \left[ \ddot{\hat{\eta}}_j(t) + \ddot{h}_i(t) - \ddot{h}_j(t) \right] + \delta_i(t). \tag{35}$$

The corresponding sliding mode vector function is defined as

$$s_i(r_i, \dot{r}_i) = \dot{r}_i(t) + K_i F_i(r_i) |r_i|^{\frac{1}{2}}, (r_i, \dot{r}_i) \in R^3 \times R^3 \tag{36}$$

where
$K_i = diag(K_{i1}, K_{i2}, K_{i3})$, $K_{ip} > 0$, $p = 1, 2, 3$, $i = 1, 2, ..., n$,
$F_i(r_i) = diag(sign(r_{i1}), sign(r_{i2}), sign(r_{i3}))$, $r_{ip} \in R$, $p = 1, 2, 3$, $i = 1, 2, ..., n$.
$|r_i|^{1/2} = [|r_1|^{1/2}, |r_2|^{1/2}, |r_3|^{1/2}]^T$.

After that, the null space of $s_i$ is the definition of the $i$th sliding surface, that is,

$$S_i(r_i, \dot{r}_i) = \left\{ (r_i, \dot{r}_i) \in R^3 \times R^3, s_i(r_i, \dot{r}_i) = 0 \right\}. \tag{37}$$

With (35) and (36), the time-varying terminal sliding mode controller is given by

$$\tau_{is}(t) = -\hat{W}_i^T \Phi_i(\hat{v}_i) - (k_2 - k_1^2) \tilde{\eta}_i(t) - k_1 \tilde{v}(t) \\ - \frac{\ell_{ij}}{\ell_{ii}} \sum \left[ \ddot{\hat{\eta}}_j(t) + \ddot{h}_i(t) - \ddot{h}_j(t) \right] \\ - \frac{1}{2} K_i z_i(r_i, \dot{r}_i) - B_i sign(s_i(r_i, \dot{r}_i)) \\ (i = 1, 2, ..., n, j = 0, 1, ..., n, j \neq i, h_0(t) = 0) \tag{38}$$

where

$(r_i, \dot{r}_i) \in M_i$, $M_i$ is a set, $z_i(\cdot, \cdot)$ is bounded,
$M_i = \left\{ (r_i, \dot{r}_i) \in R^{3 \times 1} \times R^{3 \times 1} : \|z_i(r_i, \dot{r}_i)\|_\infty \leq \lambda_i \right\}$,
$z_i(r_i, \dot{r}_i) = \left[ \dot{r}_{i1} |r_{i1}|^{-1/2}, \dot{r}_{i2} |r_{i2}|^{-1/2}, \dot{r}_{i3} |r_{i3}|^{-1/2} \right]^T$,
$\lambda_i = \|K_i\|_\infty + \gamma_i$ $(\gamma_i > 0)$,

$$sign(s_i(r_i, \dot{r}_i)) = \begin{bmatrix} sign(s_{i1}(r_i, \dot{r}_i)) \\ sign(s_{i2}(r_i, \dot{r}_i)) \\ sign(s_{i3}(r_i, \dot{r}_i)), \end{bmatrix},$$

$B_i = diag(B_{i1}, B_{i2}, B_{i3})$, $B_{ip} \in R$, $i = 1, 2, ..., n$, $p = 1, 2, 3$.

The terminal sliding mode controller is bounded because $z_i(\cdot, \cdot)$ is bounded. The necessary requirements for $M_i$ to be positively invariant with respect to (35) are presented in the following lemma.

*Lemma 5 ( [5]):* Consider the generalized error dynamics (35) with the terminal sliding mode controller (38), if $B_{ip}$ satisfy

$$B_{ip} = \beta_{ip} + \sup_{(z, \dot{r}, t) \in R^{3n} \times R^{3n} \times R} \|\delta_i(t)\|_\infty, \quad (39)$$

with

$$\beta_{ip} > \frac{\lambda_i^2 - k_{ip}\lambda_i}{2} > 0 \quad (40)$$

then the $M_i$ is positively invariant regard to (35).

Notice that the singularity issue of the terminal sliding mode controller is led by $(r_i, \dot{r}_i) \to 0$, $z_i(r_i, \dot{r}_i) \to \infty$. To avoid this issue, the above lemma is presented.

Next, to avoid the singularity problem, an auxiliary sliding surface $S_{ia}$ is designed with the corresponding controller under the initial conditions $(r_i, \dot{r}_i) \in R^n \times R^n \backslash M_i$. The auxiliary vector is designed as

$$s_{ia}(r_i, \dot{r}_i) = \dot{r}_i(t), (r_i, \dot{r}_i) \in R^3 \times R^3. \quad (41)$$

Then, the relevant sliding surface can be defined as

$$S_{ia}(r_i, \dot{r}_i) = \left\{ (r_i, \dot{r}_i) \in R^3 \times R^3, s_{ia}(r_i, \dot{r}_i) = 0 \right\}. \quad (42)$$

Applying (35) and (41) as a basis, the auxiliary sliding mode controller is proposed as

$$\begin{aligned} \tau_{isa}(t) = &-\hat{W}_i^T \Phi_i(\hat{v}_i) - (k_2 - k_1^2)\tilde{\eta}_i(t) - k_1 \tilde{v}(t) \\ &- \frac{\ell_{ij}}{\ell_{ii}} \sum \left[ \ddot{\tilde{\eta}}_j(t) + \ddot{h}_i(t) - \ddot{h}_j(t) \right] \\ &- B_{ia} sign(s_{ia}(r_i, \dot{r}_i)) \\ &(i = 1, 2, ..., n, j = 0, 1, ..., n, j \neq i, h_0(t) = 0) \end{aligned} \quad (43)$$

where $(r_i, \dot{r}_i) \notin M_i$, $B_{ia} = diag(B_{i1a}, B_{i2a}, B_{i3a})$, $B_{ipa} \in R$, $i = 1, 2, ..., n$, $p = 1, 2, 3$. Similarly,

$$B_{ipa} = \beta_{ipa} + \sup_{(z, \dot{r}, t) \in R^{3n} \times R^{3n} \times R} \|\delta_i(t)\|_\infty, \quad (44)$$

where $\beta_{ipa} > 0$, and

$$sign(s_{ia}(r_i, \dot{r}_i)) = \begin{bmatrix} sign(s_{i1a}(r_i, \dot{r}_i)) \\ sign(s_{i2a}(r_i, \dot{r}_i)) \\ sign(s_{i3a}(r_i, \dot{r}_i)) \end{bmatrix}.$$

*Remark 2:* The state space of the system is divided into two regions as there is a singularity issue. The terminal sliding mode controller (38) corresponds to the nonsingular region, and the auxiliary sliding mode controller (43) corresponds to the singular region.

*Theorem 2:* Consider the terminal sliding mode controller (38) and the auxiliary sliding mode controller (43) for an ASV team (15) and a target (12). The time-varying formation of the ASV team can be achieved while encircling the target within a finite time. The error states $(r_i, \dot{r}_i)$ can be converged to the origin within a finite time.

*Proof:* First, under the $(r_i, \dot{r}_i) \notin M_i$ condition, a Lyapunov function candidate is designed as

$$V_i(s_{ia}(r_i, \dot{r}_i)) = \frac{1}{2} s_{ia}^T(r_i, \dot{r}_i) s_{ia}(r_i, \dot{r}_i), (r_i, \dot{r}_i) \notin M_i. \quad (45)$$

Take the time differential of (45), together with (41) and (43), one has obtained:

$$\begin{aligned} \dot{V}_i(s_{ia}(r_i, \dot{r}_i)) &= s_{ia}^T(r_i, \dot{r}_i) \dot{s}_{ia}(r_i, \dot{r}_i) \\ &= s_{ia}^T(r_i, \dot{r}_i) \ddot{z}_i(t) \\ &= s_{ia}^T(r_i, \dot{r}_i)(\tau_{isa}(t) + \hat{W}_i^T \Phi_i(\hat{v}_i) \\ &+ \frac{\ell_{ij}}{\ell_{ii}} \sum \left[ \ddot{\tilde{\eta}}_j(t) + \ddot{h}_i(t) - \ddot{h}_j(t) \right] \\ &+ (k_2 - k_1^2)\tilde{\eta}_i(t) + k_1 \tilde{v}(t) + \delta_i(t)). \end{aligned} \quad (46)$$

Substituting (43) into the above equation yields

$$\begin{aligned} \dot{V}_i(t) &= s_{ia}^T(r_i, \dot{r}_i)(-B_{ia} sign(s_{ia}(r_i, \dot{r}_i)) + \delta_i(t)) \\ &\leq -\sum_{p=1}^{3} \beta_{ipa} |s_{ipa}(r_i, \dot{r}_i)| \\ &\leq -\sum_{p=1}^{3} \min_{p=1,2,3} \{\beta_{ipa}\} |s_{ipa}(r_i, \dot{r}_i)| \\ &\leq -\min_{p=1,2,3} \{\beta_{ipa}\} \|\dot{r}_i\|_1 \\ &\leq -\sqrt{2} \min_{p=1,2,3} \{\beta_{ipa}\} V_i^{\frac{1}{2}}(s_{ia}(r_i, \dot{r}_i)) \\ &(r_i, \dot{r}_i) \notin M_i, i = 1, 2, ..., n. \end{aligned} \quad (47)$$

From the Lemma 2, we can get the trajectories $(r_i, \dot{r}_i) \notin M_i$ will converge and remain in $S_{ia}(r_i, \dot{r}_i)$ within a finite time. Then, for the case $(r_i, \dot{r}_i) \in M_i$, a Lyapunov candidate function is considered as

$$V_{i1}(t) = \frac{1}{2} s_i^T(r_i, \dot{r}_i) s_i(r_i, \dot{r}_i), (r_i, \dot{r}_i) \in M_i. \quad (48)$$

Taking the time differential of $V_{i1}(t)$ yields

$$\dot{V}_{i1}(t) = s_i^T(r_i, \dot{r}_i)\dot{s}_i(r_i, \dot{r}_i)$$
$$= s_i^T(r_i, \dot{r}_i)(\ddot{z}_i(t) + \frac{1}{2}K_i z_i(r_i, \dot{r}_i))$$
$$= s_i^T(r_i, \dot{r}_i)(\tau_{is}(t) + \hat{W}_i^T \Phi_i(\hat{v}_i) + (k_2 - k_1^2)\tilde{\eta}_i(t)$$
$$+ k_1\tilde{v}(t) + \frac{\ell_{ij}}{\ell_{ii}}\sum\left[\ddot{\tilde{\eta}}_j(t) + \ddot{h}_i(t) - \ddot{h}_j(t)\right]$$
$$+ \frac{1}{2}K_i z_i(r_i, \dot{r}_i) + \delta_i(t)). \tag{49}$$

Substitute (38) into (49), one has obtained

$$\dot{V}_{i1}(t) = s_i^T(r_i, \dot{r}_i)(-B_i sign(s_i(r_i, \dot{r}_i)) + \delta_i(t))$$
$$\leq -\sum_{p=1}^{3}\beta_{ip}\left|s_{ip}(r_i, \dot{r}_i)\right|$$
$$\leq -\sum_{p=1}^{3}\min_{p=1,2,3}\{\beta_{ip}\}\left|s_{ip}(r_i, \dot{r}_i)\right| \tag{50}$$
$$\leq -\min_{p=1,2,3}\{\beta_{ip}\}\|\dot{r}_i\|_1$$
$$\leq -\sqrt{2}\min_{p=1,2,3}\{\beta_{ip}\}V_i^{\frac{1}{2}}(s_i(r_i, \dot{r}_i))$$
$$(r_i, \dot{r}_i) \in M_i, i = 1, 2, ..., n.$$

From the Lemma 2, we can get the trajectories of $(r_i, \dot{r}_i) \in M_i$ will converge to $S_i(r_i, \dot{r}_i)$ and stay on this surface within a finite time. To sum up, the error states $(r_i, \dot{r}_i)$ are able to converge to the sliding surface and remain there within a finite time, regardless of the beginning states. Additionally, as indicated by (36), while on the sliding surface $S_i$, the dynamics of the closed-loop error are described by

$$\dot{r}_i(t) = -K_i F_i(r_i)|r_i|^{\frac{1}{2}}, r_i \in S_i. \tag{51}$$

Then, the Lyapunov function is considered as

$$V_i(r_i) = \|r_i\|_1. \tag{52}$$

Taking the time differential of $V_i(r_i)$ yields:

$$\dot{V}_i(r_i) = \sum_{p=1}^{3}sign(r_{ip})\dot{r}_{ip}$$
$$= -\sum_{p=1}^{3}k_{ip}|r_{ip}|^{\frac{1}{2}} \tag{53}$$
$$\leq -\left\|K_i^{-1}\right\|_\infty^{-1}(V_i(r_i))^{\frac{1}{2}}$$
$$r_i \in S_i, i = 1, 2, ..., n$$

Based on Lemma 2 with $\Omega_0 = \{0\}$, the error states (51) can converge to zero within a finite time. To sum up, according to Lemma 4, the ASV team can achieve the encircle motion and predefined time-varying formation within a finite time. ∎

TABLE I
THE PARAMETERS OF EACH ASV

| Elements | Values | Elements | Values | Elements | Values |
|---|---|---|---|---|---|
| $I_z$ | 1.70 | $Y_{|z|z}$ | -2.00 | $J_{|y|z}$ | -4.00 |
| $Y_z$ | 0.10 | $J_{|y|y}$ | 5.00 | $J_{\dot{z}}$ | -1.00 |
| $x_g$ | 0.04 | $Y_{|y|y}$ | -36.00 | $J_{|z|z}$ | -4.00 |
| $X_{xxx}$ | -5.80 | $J_y$ | 0.10 | $Y_{\dot{z}}$ | -0.00 |
| $X_x$ | -0.72 | $Y_{|y|z}$ | 2.00 | $X_{\dot{x}}$ | -2.00 |
| $Y_y$ | -0.86 | $J_z$ | -6.00 | $J_{\dot{y}}$ | -0.00 |
| $X_{|x|x}$ | -1.30 | $Y_{|z|y}$ | -3.00 | $Y_{\dot{y}}$ | -10.00 |

## IV. SIMULATION EXAMPLE

According to the specifications detailed in [5], it is assumed that the parameters of all ASVs in this simulation are uniform. The parameters are shown in Table I. This simulation example reproduces the strategy scenario of dynamic encirclement for the multi-ASV system to test the proposed approach. Finally, the efficiency of the developed time-varying sliding mode controller under saturated actuators and unavailable velocity state information is validated by the simulation results.

In this numerical simulation, an ASV team consisting of four ASVs and a Target ASV are considered. The initial states of the target ASV at $0s$ are set as $\eta_0(0) = [0, 2, \pi/4]^T$, $v_0(0) = [3, -3, 0]^T$. The initial states of the ASV team include $\eta_1(0) = [-5, 5, \pi/4]^T$, $v_1(0) = [1, 1, 0]^T$, $\eta_2(0) = [-5, 10, -\pi/4]^T$, $v_2(0) = [1, 0.5, 0.25]^T$, $\eta_3(0) = [-5, 0, \pi/2]^T$, $v_3(0) = [0.5, 1, -0.3]^T$, $\eta_4(0) = [-5, -5, 0]^T$, $v_4(0) = [1, 0.5, 0.2]^T$. The preset time-varying formation function is given by:

$$h_i(t) = \begin{bmatrix} 5\cos(0.1t + 2\pi(i-1)/4) \\ 5\sin(0.1t + 2\pi(i-1)/4) \\ 0.1 \end{bmatrix}.$$

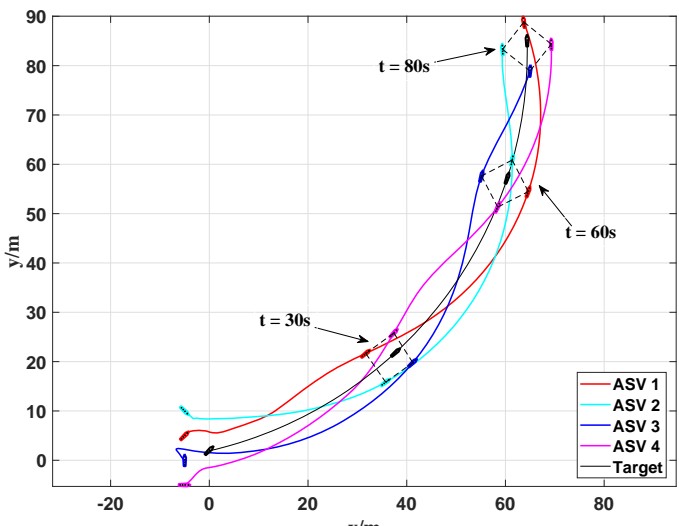

Fig. 2. Trajectories of the ASV team and one target.

The controller parameter $K_i$ is set as $K_i = diag(3, 3, 3)$ and the control gains are set as $B_i = B_{ia} = diag(2, 2, 2)$.

The gain matrix of the adaptive fuzzy state observer is set as $k_1 = 5$, $k_2 = 10$. The adaptive law parameters of a FLS are set as $\Lambda_i = 3$, $\alpha_i = 2$. The directed Laplacian matrix $L$ is given by

$$L = \begin{bmatrix} 0 & 0 & 0 & 0 & 0 \\ -\frac{1}{4} & 1 & -\frac{1}{4} & -\frac{1}{4} & -\frac{1}{4} \\ -\frac{1}{4} & -\frac{1}{4} & 1 & -\frac{1}{4} & -\frac{1}{4} \\ -\frac{1}{4} & -\frac{1}{4} & -\frac{1}{4} & 1 & -\frac{1}{4} \\ -\frac{1}{4} & -\frac{1}{4} & -\frac{1}{4} & -\frac{1}{4} & 1 \end{bmatrix}. \quad (54)$$

Fig. 2 shows the trajectories of the ASV team and the target ASV. The ASV team completed the dynamic encirclement of the target ASV while also achieving the time-varying formation. Besides, the four sets of ASVs in Fig. 2 represent the positions between the ASV team and the target ASV at $0s$, $30s$, $60s$, and $80s$, respectively. By observing the three connected markers, it can be indicated that the predefined time-varying formation is a rotating quadrilateral.

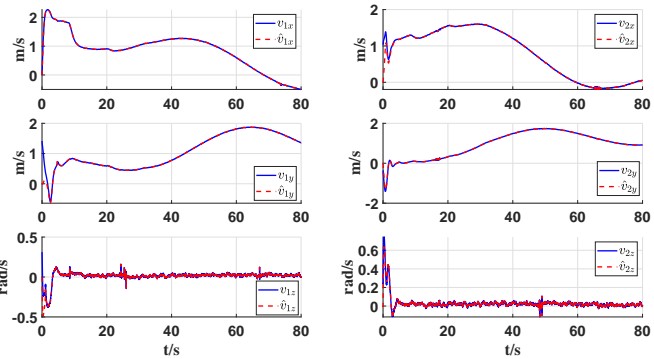

Fig. 3. A comparison of the estimated velocities and actual velocities of ASV 1 and ASV 2.

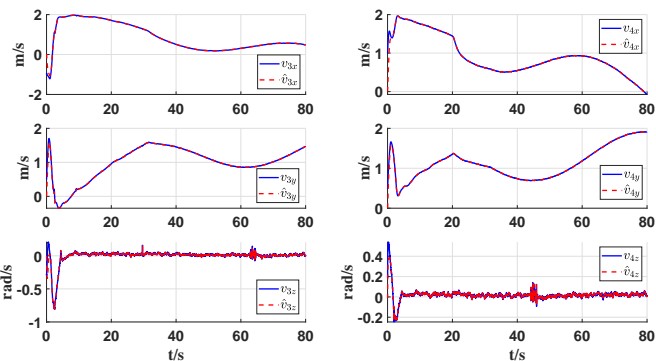

Fig. 4. A comparison of the estimated velocities and actual velocities of ASV 3 and ASV 4.

Since this paper considers the case of unmeasurable velocities, an adaptive fuzzy state observer is implemented. Fig. 3 and Fig. 4 compare the observed and actual velocities of each ASV to demonstrate the capability of the developed adaptive fuzzy state observer. At the beginning, the observed velocities for all four ASVs are zero. Gradually, the observed velocity curves align closely with the actual velocity curves. The state

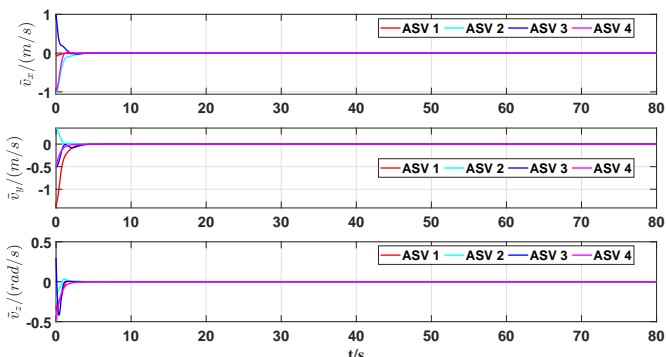

Fig. 5. The observation error of the velocities.

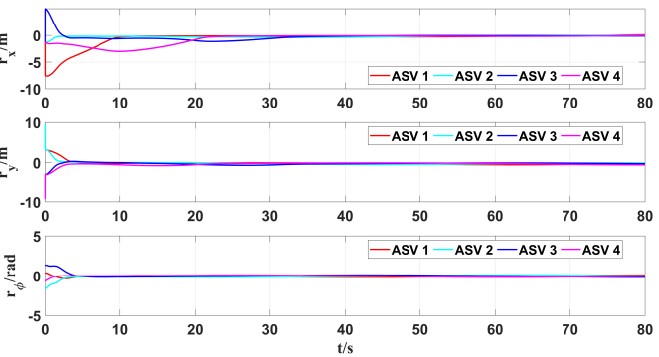

Fig. 6. The generalized error states.

observer capability for ASV 1 and ASV 2 is shown in Fig. 3, while that for ASV 3 and ASV 4 is indicated in Fig. 4. It can be illustrated that, even with unknown initial velocities, the observer quickly estimates the actual velocities within a short time.

Fig. 5 shows the observation error of the velocities estimated by the designed observer. It can be indicated that the velocity estimation errors eventually converge to zero. Together with Fig. 3 and Fig. 4, the efficiency of the developed adaptive fuzzy state observer is demonstrated in situations where velocity is not measurable.

The generalized error states are shown in Fig. 6. As shown

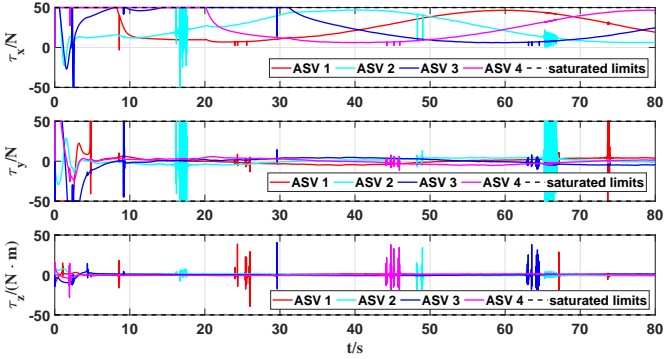

Fig. 7. Saturated inputs of the ASV team.

in the figure, the generalized error states of the ASV team, despite unmeasured velocity states, saturated inputs, and external disturbances, ultimately converge to zero within a finite time under the designed sliding mode controller.

Fig. 7 indicates the control inputs calculated by the developed time-varying sliding mode controller. Notice that the dashed lines represent the saturated limits of the control inputs, and the real lines represent the saturated control inputs. The control inputs are limited by

$$\begin{cases} -50N \leq \tau_x \leq 50N \\ -50N \leq \tau_y \leq 50N \\ -50N \cdot m \leq \tau_z \leq 50N \cdot m. \end{cases} \quad (55)$$

## V. Conclusion

The finite-time dynamic encirclement issues are investigated for multi-ASV systems with unavailable velocities using time-varying sliding mode control in this paper. The multi-ASV system suffers from saturated inputs and external disturbances. First, the nonlinear dynamics are approximated by a FLS, and an adaptive fuzzy state observer is developed to estimate the ASVs velocities. In addition, the effectiveness of the developed state observer is verified using Lyapunov stability theory. Then, based on the finite-time theory, a distributed time-varying formation sliding mode controller is proposed to achieve encircle motion and time-varying formation within a finite time. Finally, Lyapunov theory is utilized to verify the stability of the closed-loop system. A series of simulation examples are presented to demonstrate the efficiency of the developed finite-time dynamic encirclement approach.

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
