# OpenReview forum: "Observer-Based Finite-Time Dynamic Encirclement for Multi-ASV Systems Using Time-Varying Sliding Mode Control"
_IEEE.org/ICIST/2024/Conference — IEEE ICIST 2024 Conference Submission_

### Official Review · Reviewer_x4FD · 2024-08-21
**Accpet**

**Rating:** 7
**Confidence:** 5

**Review:**

This paper investigates the finite-time dynamic encirclement issues for multi-Autonomous Surface Vehicle (ASV) systems with unavailable velocities using time-varying sliding mode control. The multi-ASV system faces challenges such as saturated inputs and external disturbances.However, there are opportunities for improvement in the following aspects:1) The contributions should be stressed more in comparison with specific existing works.2)The impact of controller parameters on system performance should be explored, and parameter tuning rules should be conducted to further enhance system performance.

---

### Official Review · Reviewer_GxUA · 2024-08-23
**This is a good paper.**

**Rating:** 8
**Confidence:** 5

**Review:**

This paper proposes a novel approach to handle finite-time dynamic encirclement problems for multiautonomous surface vehicle (ASV) systems. A fuzzy logic system (FLS) is utilized to approximate the nonlinear dynamics, and an adaptive fuzzy state observer is designed to estimate the unavailable velocities of the multi-ASV system. Finally, the effectiveness of the obtained results can be verified by the Lyapunov theory and simulation examples presented.
1). Please elaborate the difference of the sliding surface (37) and the sliding surface (42).
2). The main motivation of the proposed scheme should be described more clearly.

---

### Official Review · Reviewer_BSMo · 2024-08-25
**The manuscript is well-organized and clearly stated**

**Rating:** 9
**Confidence:** 4

**Review:**

In the manuscript titled"Observer-Based Finite-Time Dynamic Encirclement for Multi-ASV Systems Using Time-Varying Sliding Mode Control"introduces a novel method for finite-time dynamic encirclement of multi-autonomous surface vehicles, employing fuzzy logic, adaptive observers, and sliding mode controllers to address model uncertainties and external disturbances, with validation through Lyapunov theory and simulations. This work provides new insights into the development of finite-time dynamic encirclement strategies for multi-ASV systems by leveraging fuzzy logic and sliding mode control to handle challenges such as unavailable velocities and external disturbances.The manuscript is well-organized and clearly stated.And its motive is reasonable, the literature contrast is sufficient, the innovation point is bright.

---

### Decision · Program_Chairs · 2024-09-08

Accept (Oral)